# Physiology-informed regularisation enables training of universal differential equation systems for biological applications

Max de Rooij [1,2]*, Balázs Erdős [3], Natal A. W. van Riel [1,2,4], Shauna D. O'Donovan [1,2]

1 Department of Biomedical Engineering, Eindhoven University of Technology, Eindhoven, Netherlands, 2 Eindhoven Artificial Intelligence Systems Institute (EAISI), Eindhoven University of Technology, Eindhoven, Netherlands, 3 Department of Data Science and Knowledge Discovery, Simula Metropolitan Center for Digital Engineering, Oslo, Norway, 4 Amsterdam University Medical Center, University of Amsterdam, Amsterdam, Netherlands

* m.d.rooij@tue.nl

**Data availability statement:** The source code and data used to produce the results and

## Abstract

Systems biology tackles the challenge of understanding the high complexity in the internal regulation of homeostasis in the human body through mathematical modelling. These models can aid in the discovery of disease mechanisms and potential drug targets. However, on one hand the development and validation of knowledge-based mechanistic models is time-consuming and does not scale well with increasing features in medical data. On the other hand, data-driven approaches such as machine learning models require large volumes of data to produce generalisable models. The integration of neural networks and mechanistic models, forming universal differential equation (UDE) models, enables the automated learning of unknown model terms with less data than neural networks alone. Nevertheless, estimating parameters for these hybrid models remains difficult with sparse data and limited sampling durations that are common in biological applications. In this work, we propose the use of physiology-informed regularisation, penalising biologically implausible model behavior to guide the UDE towards more physiologically plausible regions of the solution space. In a simulation study we show that physiology-informed regularisation not only results in a more accurate forecasting of model behaviour, but also supports training with less data. We also applied this technique to learn a representation of the rate of glucose appearance in the glucose minimal model using meal response data measured in healthy people. In that case, the inclusion of regularisation reduces variability between UDE-embedded neural networks that were trained from different initial parameter guesses.

analyses presented in this manuscript are available from a GitHub repository at https://github.com/Computational-Biology-TUe/ude-regularization. We have also used Zenodo to assign a DOI to the repository: https://doi.org/10.5281/zenodo.11402365

**Funding:** The research presented in this manuscript was supported by a Starting Package from the Eindhoven AI Systems Institute (EAISI) awarded to S. O'D. N.A.W.v.R. is supported by a grant from the Dutch Research Council (NWO) [https://www.nwo.nl/] as part of the Complexity Program (project number 645.001.003). The funders had no role in study design, data collection and analysis, decision to publish, or preparation of the manuscript. All other authors received no specific funding for this work.

**Competing interests:** The authors have declared that no competing interests exist.

## Author summary

Systems biology concerns the modelling and analysis of biological processes, by viewing these as interconnected systems. Modelling is typically done either using mechanistic differential equations that are derived from experiments and known biology, or using machine learning on large biological datasets. While mathematical modelling from biological experiments can provide useful insights with limited data, building and validating these models takes a long time and often requires highly invasive measurements in humans. Efforts to combine this classical technique with machine learning have resulted in a framework termed universal differential equations, where the model equations contain a neural network to describe unknown biological interactions. While these methods have shown success in numerous fields, applications in biology are particularly challenging due to limited data-availability and high data sparsity. In this work, we have introduced physiology-informed regularisation to overcome these instabilities and to constrain the model to biologically plausible behaviour. Our results show that by using physiology-informed regularisation, we can more accurately predict future unseen observations in a simulated example, when trained with much more limited data than a similar model without regularisation. Additionally, we show an application of this technique on human data, applying a neural network to learn the appearance of glucose in the blood plasma after a meal.

## Introduction

The high complexity of the internal regulation of homeostasis in the human body, combined with increasing levels of detail of biological datasets, often makes direct interpretation of measurements difficult. In systems biology, this challenge is approached through dynamic mathematical modelling, which aims to systematically combine known biology and hypotheses with quantitative data [1]. An advantage of representing a biological system as a set of mathematical equations is that simulations can be performed rapidly using numerical methods providing insights into how individual interactions can give rise to systemic behavior over time and allowing for *in silico* hypothesis testing.

The model development cycle, whereby models are constructed, iteratively refined, and validated by comparison with experimental data, provides a practical basis for computer-assisted biology research. In this way, biological interactions which are difficult to isolate in *in vitro* experiments can be uncovered [2,3]. For example, a recent model revealed the context-specific mechanism by which the hypoxia-inducible factor $1\alpha$ (HIF1$\alpha$) regulates lipid-accumulation [4] and a model of the liver X-receptor (LXR) revealed that the origin of LXR-induced hepatic steatosis lies in an increase of the free fatty-acid influx in the liver [5]. However, the model development cycle is highly laborious and time consuming, and larger dimensionalities of data further complicate the construction of reliable mathematical models [1].

With the introduction of high-throughput sequencing technology, the number of features in medical data is growing quickly. However, the collection of data at the time resolution and duration necessary to train many commonly used machine learning methods is often infeasible in practice. Furthermore, while the generation of data with wearable devices achieves sufficient temporal resolution, they are still limited to specific features, such as continuous glucose monitoring or physical activity monitoring with smartwatches. A promising solution is to integrate previously built mechanistic models, rooted in known biology, with highly

flexible machine learning approaches. During training, the mechanistic model simulates the known biology underlying the system, allowing the machine learning model to learn only the unknown relationship from the data. This technique is labeled Universal Differential Equation (UDE) modeling, referring to the universal approximation property [6] of neural networks [7,8]. These universal differential equations have been shown to recover missing interactions in examples from materials science, astrophysics, and epidemiology, and have been successfully applied in environmental sciences, capturing missing terms in glacier ice flows and climate models [9,10].

The introduction of a neural network introduces a large number of parameters into an already complex non-linear biological model and therefore heavily influences the already complicated loss landscape of a parameterised dynamic model. When combined with limited data availability due to practical limitations of collecting data from humans, this presents itself as an increase in local minima in the loss function, having the potential to negatively impact the ability of the model to find a generalisable solution.

This challenge of training a universal differential equation system in situations where limited data is available is an active area of research. An approach by Roesch et al. proposed a two-stage training approach to reduce the probability of a model getting stuck in a local minimum [11]. However, because this method initially relies on splines of the data, it is sensitive to low sampling frequencies as often encountered in biomedical data [12]. Additionally, data needs to be available for all state variables in the model, which is rarely the case in biological systems. Vortmeyer-Kley et al. instead suggested a modification to the loss function, minimising the angle between the predicted and given points in the state-space of the system, in addition to the difference between the two points [13]. Additionally, Turan and Jaschke proposed splitting the data into smaller parts and computing the loss of the ODE system for each part separately. This approach is labeled multiple shooting [14]. Even though these approaches can lead to improvements in neural network behavior within differential equation systems, these solutions depend on having either a sufficient sampling frequency or a sufficient sampling duration, and do not take advantage of the wealth of knowledge available for many of these systems.

Instead, we propose to constrain the model solution to physiologically plausible regions by incorporating this additional biological knowledge in a simple fashion through physiology-informed regularisation. In a general sense, regularisation is a technique in machine learning that aims to reduce overfitting. In this study, physiology-informed regularisation is implemented as a form of Tikhonov regularisation, where the cost function is supplemented with terms penalising biologically undesirable behaviours, such as negative metabolite concentrations or metabolites disappearing or being created out of nothing. In addition to reducing overfitting of the neural network, this form of regularisation promotes the identification of biologically plausible behaviours in these larger systems. The type of biological information that is included in this way can vary per system, depending on the knowledge of the system, the available data, and the specific term that the neural network is representing. This type of physiology-informed regularisation has been applied previously in classical ODE modelling [3,15], and a similar technique was recently used in an epidemiological model using the UDE approach on densely sampled data [16]. However, the impact of physiology-informed regularisation on the training process in situations of varying data sparsity and sampling time, as well as varying the strength of different regularisation terms has not been investigated previously in universal differential equation models.

In this work, we implement simple forms of physiology-informed regularisation to aid in the training of two UDE systems. Firstly, we add a penalty to prevent negative concentrations in the context of a simple model, learning the conversion of one molecule to another via

Michaelis-Menten kinetics and demonstrate that physiology-informed regularisation supports the recovery of more correct biological interactions. Additionally, we investigate the effect of physiology-informed regularisation in scenarios with varying sampling duration and scarcity. To demonstrate the benefit of physiology-informed regularisation beyond a simulated example, we also applied it to a second scenario, learning the rate of appearance of glucose from a meal from human data in the glucose minimal model, by Bergman [17], where the UDE term is constrained to be non-negative and have an area under the curve of glucose influx in the blood plasma that corresponds to the amount of ingested glucose.

## Materials and methods

### General structure of a UDE model

For state-variables $\mathbf{u}(t)$, a universal differential equation (UDE) can be formulated as:

$$\frac{d\mathbf{u}(t)}{dt} = f(\mathbf{u}(t), \mathbf{p}, t) + N(\mathbf{u}(t), \boldsymbol{\theta}, t) \tag{1}$$

Here, $f(\mathbf{u}(t), \mathbf{p}, t)$ represents the known terms in the differential equation model describing the state variables $\mathbf{u}(t)$, over the time $t$ and possibly *a priori* unknown parameters $\mathbf{p}$.

In a typical parameter estimation procedure, these parameters $\mathbf{p}$ are estimated from data by minimising the difference between the data and model predictions using numerical optimisation techniques. The neural network, shown in equation 1 as $N(\mathbf{u}(t), \boldsymbol{\theta}, t)$, where $\boldsymbol{\theta}$ indicates the neural network parameters, represents unknown terms. Similarly to the conventional parameter estimation procedure, the parameters of the UDE model are estimated from data using numerical optimisation. Both $\mathbf{p}$ and $\boldsymbol{\theta}$ can be estimated simultaneously [18]. This formulation differs from other hybrid modelling techniques, such as a physics-informed neural network (PINN), where the dynamics of the state-variables is described entirely using a neural network [19]. Additionally, note that the formulation of a UDE in equation 1 is for illustrative purposes. The exact formulation of a UDE is not limited to a summation of the analytical terms and a neural network. Another possibility includes the multiplication of an analytical term with a neural network.

### Regularisation on simulated data of Michaelis-Menten kinetics

To be able to determine the effect of regularisation on the ability of the UDE model to recover unknown interactions, a simulation study was performed. By simulating data from a known model, the error with respect to the true model for future unseen data can be determined.

**Model description.**  In this simulation, the known model consists of two ordinary differential equations (ODEs) coupled by a saturable enzymatic conversion term implemented as a Michaelis-Menten equation, which is commonly found in models of biological pathways [20]. The Michaelis-Menten model used here can be described mathematically as

$$\frac{dS(t)}{dt} = k_S S(t) - k_{SP} \frac{S(t)}{k_M + S(t)} \tag{2}$$

$$\frac{dP(t)}{dt} = k_{SP} \frac{S(t)}{k_M + S(t)} - k_P P(t) \tag{3}$$

Where $S(t)$ and $P(t)$ represent the concentrations (in mM) of two distinct metabolites or molecules, $P(t)$ is produced from $S(t)$ through an enzymatic conversion with rate constant $k_{SP}$ and Michaelis-Menten coefficient $k_M$. Furthermore, metabolite $S(t)$ stimulates its own

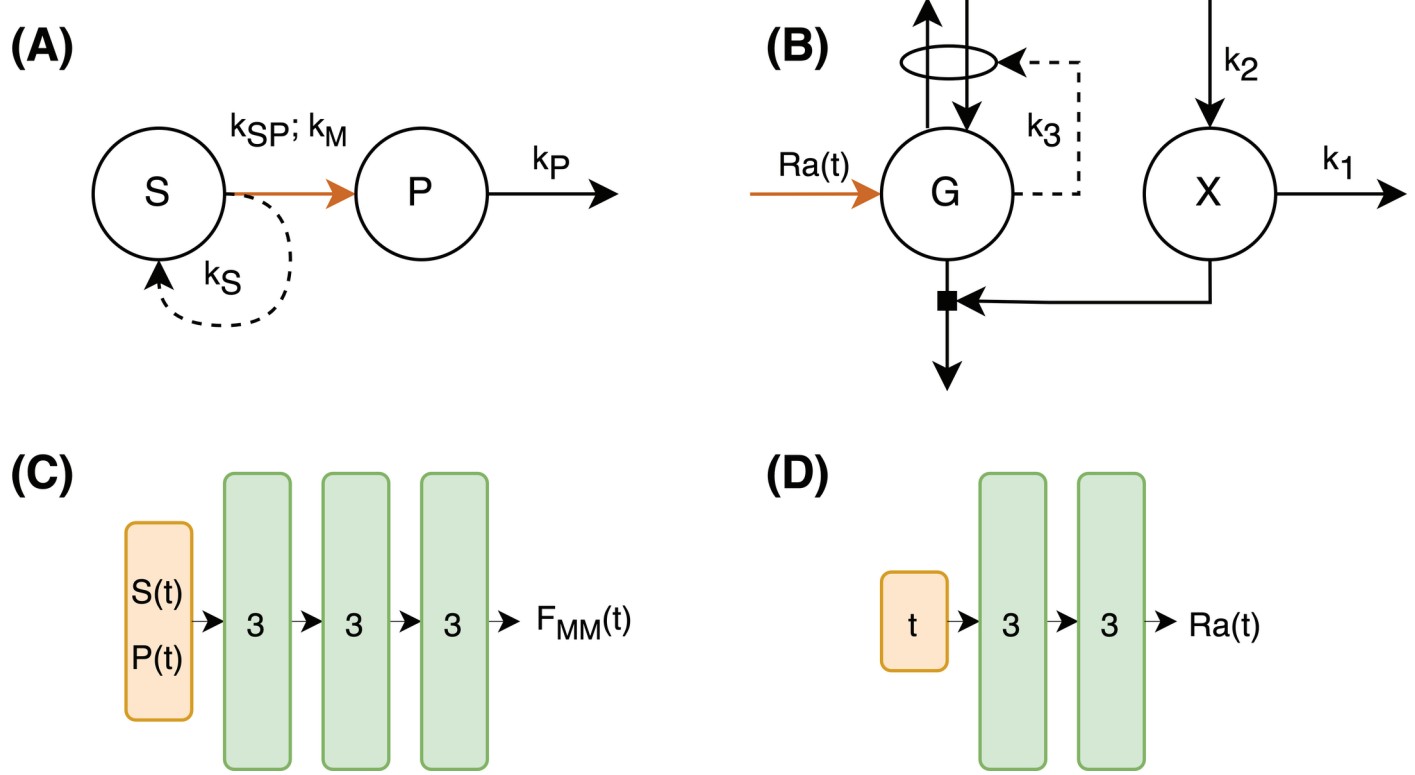

**Fig 1. Overview of implemented model structures and architectures of incorporated networks.** A: Graphical representation of the simulated model with Michaelis-Menten kinetics. B: Graphical representation of the glucose minimal model. Interactions are given using dashed arrows, while fluxes are shown in solid arrows. The location of the neural network in both models is indicated by the orange arrow. C: Neural network architecture for the Michaelis-Menten model with inputs $S(t)$ and $P(t)$ and output $F_{\mathrm{MM}}(t)$, which is the flux for the conversion from $S(t)$ to $P(t)$. D: Neural network architecture for the glucose minimal model with input $t$ and output $Ra(t)$. All layers are densely connected.

production through rate parameter $k_S$, and metabolite $P(t)$ decays linearly with its concentration and the decay rate $k_P$, displayed graphically in Figure 1A. All parameters and initial conditions for the ground truth model simulation are described in Table 1. The time variable was taken to be measured in minutes. This and all other models in this work were implemented and simulated using Julia version 1.10 [21] and the Tsit5-solver [22] in the 'DifferentialEquations' package [18].

**Data description.** Data was simulated using the model and parameters described in Table 1. The simulated data was split into high-resolution validation data, sampled every

**Table 1. Parameter values and initial state values used for data simulation of the Michaelis-Menten model.**

| Parameter | Value | Unit |
|---|---|---|
| $k_S$ | 0.05 | $\mathrm{min}^{-1}$ |
| $k_P$ | 0.08 | $\mathrm{min}^{-1}$ |
| $k_{SP}$ | 0.2 | $\mathrm{mM} \cdot \mathrm{min}^{-1}$ |
| $k_M$ | 1.1 | $\mathrm{mM}$ |
| $S_0$ | 2 | $\mathrm{mM}$ |
| $P_0$ | 0 | $\mathrm{mM}$ |

0.1 minutes between 0 and 400 minutes, and several lower resolution training datasets. Each training dataset represents a combination of values set for the sampling frequency and the duration of sampling. Samples were extracted at intervals of 5, 10, or 20 minutes for sampling durations of 20 to 100 minutes. Additionally, noise was added to each training dataset to simulate measurement conditions. The noise distribution was assumed to be normal, with a standard deviation of 5% of the maximum value in the validation dataset of the corresponding state-variable. This results in a dataset with a minimum noise of 5%, similar to an earlier study of hybrid modelling on concentration data [23] Additionally, we specifically tested one case, with sampling time of 40 minutes 5 minutes between samples with 10% noise. In the cases where addition of noise resulted in negative measurements, measurement values were set to 0.

**Construction and training of the UDE model.** The UDE model is formed by removing the Michaelis-Menten coupling term and replacing it with a single fully connected neural network with the two state variables $S(t)$ and $P(t)$ at time $t$ as inputs, three hidden layers of size 3 with Gaussian radial basis activation functions, and a single output with linear activation representing the flux of the conversion between $S(t)$ and $P(t)$(Figure 1C). This structure was determined based on a grid search of neural network structures with layer widths 2 to 4 and network depths 1 to 3. The data used for the grid search contained samples every 5 minutes up to 100 minutes. The Gaussian radial basis activation function can be formulated as

$$\mathrm{rbf}(x) = e^{-x^2} \tag{4}$$

In this case, the parameter values for $k_S$ and $k_P$ were assumed to be known and fixed to the values in Table 1. For implementation of the neural network, the Lux.jl package [24] was used.

Using this implementation, a loss function for estimating the neural network parameters was formulated as

$$\mathrm{Loss}\left(\theta|\mathbf{u}^{\mathrm{obs}}\right) = \left\|\mathbf{u}^{\mathrm{mod}}\left(\theta\right) - \mathbf{u}^{\mathrm{obs}}\right\|_2 + \lambda \cdot R\left(\theta\right) \tag{5}$$

This equation then indicates the sum of squared differences between model predictions at the sample measurement times ($\mathbf{u}^{\mathrm{mod}}\left(\theta\right)$) for neural network parameters $\theta$ and data ($\mathbf{u}^{\mathrm{obs}}$), combined with a regularisation function $R\left(\theta\right)$, of which its contribution is controlled by a scaling hyperparameter $\lambda$. As such, the case where $\lambda = 0$, represents an unregularised setting. In this instance, the regularisation function was chosen to prevent the neural network from predicting negative concentrations of either state variables $S(t)$ or $P(t)$, resulting in a one-sided penalty term formulated as

$$R\left(\theta\right) = \left\|\min\left(\mathbf{u}^{\mathrm{mod}}\left(\theta\right); \mathbf{0}\right)\right\|_2 \tag{6}$$

In this formulation, any positive value for the model predicted concentration remains unpenalised, while the penalty on negative concentrations grows quadratically with their magnitudes.

Estimation of neural network parameters was achieved by performing gradient-based optimisation, starting from a random initial set of parameters, to find a parameter set that minimises the loss function. To improve convergence, a two-stage optimisation procedure was used, starting out with the Adam [25] optimiser with a learning rate of 0.01 for 500 iterations, followed by the Broyden-Fletcher-Goldfarb-Shanno (BFGS) [26–29] algorithm in the second

stage. The BFGS algorithm was used with an initial step norm of 0.01, running until convergence based on a function tolerance of 1e-6 and an input tolerance of 1e-6, or until the maximum number of iterations, set at 1000, was reached. The optimisation procedure was carried out in Julia version 1.10 using the 'Optimization' package [30].

For each simulated training dataset, optimisation was performed 100 times using random initialisation of the neural network weights. This procedure was carried out using regularisation strengths of 0 (no regularisation), and $10^x$ for $x \in \{-9, -8, -7, -6, -5, -4, -3, -2, -1, 0\}$.

**Evaluation of regularisation in simulated conditions.**   To investigate the effect of the regularisation penalty, model performance was evaluated by computing an evaluation error based on the densely sampled simulated validation dataset ($\mathbf{u}^{gt}$) as

$$\text{Err}_{val} = \log_{10}\left(\frac{1}{n}\left\|\mathbf{u}^{mod} - \mathbf{u}^{gt}\right\|_2\right) \tag{7}$$

The value of $n$ represents the number of validation points, which was set to a value of 4001, corresponding to the amount of points in the generated validation dataset.

## Effect of physiology-based regularisation on human data.

In addition to the simulation study, the effect of physiology-based regularisation on the training of a UDE model was also evaluated in a real-life setting. As a test case, a neural network was introduced into the glucose minimal model [17,31] to learn a representation of the rate of glucose appearance from the gut, following the ingestion of a meal and was trained using a large public dataset of meal responses.

**Model description.**   The glucose minimal model describes the dynamics of plasma glucose regulation by insulin at a course-grained level and is formulated as

$$\frac{dG(t)}{dt} = -G(t)X(t) - k_3\left(G(t) - G_0\right) + \frac{f_G M}{V}\text{Ra}(t) \tag{8}$$

$$\frac{dX(t)}{dt} = -k_1 X(t) + k_2\left(I(t) - I_0\right) \tag{9}$$

Where $G$ and $X$ describe plasma glucose concentration and a generic glucose lowering effect respectively. The model has two forcing functions, $\text{Ra}(t)$ describing the appearance of glucose from a meal in the plasma, and $I(t)$ for incorporating the measured plasma insulin values. $G_0$ and $I_0$ denote the measured fasting glucose and insulin concentrations respectively, which establish the set-points of the model. $M$ represents the mass of glucose in the meal in milligrams (mg), $f_G$ is a conversion factor from mg to mmol glucose, and $V$ is the distribution volume of glucose in liters, computed from the body weight as in [15] using

$$V = 0.26\sqrt{70 \cdot \text{BW}} \tag{10}$$

Where BW refers to the body weight in kg. The values of $M$, $f_G$, and $V$ are provided in Table 2. The model has three kinetic parameters, $k_1$, $k_2$, and $k_3$ which describe the rate of decay of $X$, the production of $X$ as a function of the plasma insulin measurement, and the rate of a general insulin-independent plasma glucose removal, respectively. A visual representation of the model is shown in Figure 1B.

**Data description.**   To train this model, the meal response data for 1,002 individuals from the PREDICT UK study were used [32]. Population demographics are shown in Table 3. The data contained plasma glucose and insulin concentrations measured at 0, 15, 30, 60, 120, 180,

**Table 2. Parameter and constant values used for the glucose minimal model.**

| Variable | Value | Unit | Description |
|---|---|---|---|
| $f_G$ | $5.551 \cdot 10^{-3}$ | $mmol \cdot mg^{-1}$ | Conversion of glucose from mg to mmol |
| $M$ | $85.5 \cdot 10^3$ | mg | Mass of glucose in the meal in mg |
| $V$ | 18.57 | L | Volume of distribution of glucose in the blood plasma |
| $k_1$ | $4.91 \cdot 10^{-2}$ | $min^{-1}$ | Rate of decay of insulin action |
| $k_2$ | $2.75 \cdot 10^{-5}$ | $mL \cdot \mu IU^{-1} \cdot min^{-1}$ | Production of insulin action based on plasma insulin |
| $k_3$ | $4.61 \cdot 10^{-2}$ | $min^{-1}$ | Rate of insulin independent glucose removal |

**Table 3. Population demographics of the PREDICT UK cohort.**

| Variable (unit) | Value |
|---|---|
| Sex (M/F) | 279/729 |
| Age (Years) | $45.48 \pm 11.88$ |
| Body weight (kg) | $72.88 \pm 15.27$ |
| Height (cm) | $168.53 \pm 10.51$ |
| BMI (kg m$^{-2}$) | $25.59 \pm 5.02$ |
| Fasting plasma glucose (mM) | $4.96 \pm 0.48$ |
| Fasting plasma insulin (mU L$^{-1}$) | $6.13 \pm 4.27$ |

and 240 minutes following the consumption of a standardised solid meal. The consumed meal contained 85.5 g of carbohydrates, 52.7 g of fat and 16.1 g of protein. Individuals with missing values for glucose or insulin in any time point for this meal were removed. Data from the remaining 903 subjects is averaged across individuals for each time point. This average meal response data is used to train the UDE-glucose model.

**Construction and training of the UDE model.** In this instance, the UDE approach is deployed to estimate the plasma glucose appearance ($Ra(t)$) in the glucose minimal model. Only the rate-of-appearance function is estimated from data, while values for the model parameters $k_1$, $k_2$, and $k_3$ are assumed to be known. To obtain values for the parameters $k_1$, $k_2$, and $k_3$, an initial rate-of-appearance function was assumed using a mechanism described by Korsbo et al. [33] (see S2 Text). $k_1$, $k_2$, $k_3$ were estimated through least-squares estimation on the PREDICT data and subsequently fixed to the values depicted in Table 2.

The same neural network structure as in the Michaelis-Menten model was taken, but the network depth was reduced by one, because of the lower number of network inputs. This resulted in a fully connected network with time $t$ as its only input, two hidden layers of size 3 with Gaussian rbf activations (equation 4) and one output layer with linear activation (Figure 1D). This network was used to estimate $Ra(t)$. In addition to the nonnegativity regularisation, used in the simulated Michealis-Menten condition, an additional regularisation penalty was incorporated to the training of the glucose-minimal UDE to ensure the area under the rate-of-appearance curve should equal 1. This penalty ensures that all the glucose from the provided meal would be taken up into the plasma compartment of the model. The regularisation penalty was formulated as in equation 11, with 480 min, to ensure the model describes full glucose absorption within 8 hours after meal consumption. During model training, the integral was approximated using the trapezoid rule with a time step of 1 min.

$$R_{auc}(\theta) = \left\| \int_0^{480} Ra(t;\theta)dt - 1 \right\|_2 \tag{11}$$

The values for regularisation strengths were set to either 0 (no regularisation) or a value in $10^x$ with $x \in \{-2, -1, 0, 1, 2\}$ for both types of regularisation. Optimisation was performed for all possible combinations of both the nonnegativity regularisation and the area-under-curve regularisation. The neural network parameters were initialised with random parameters sampled from a zero mean Gaussian distribution with a standard deviation of $1 \cdot 10^{-4}$. For each experimental condition, 100 initialisations were used.

## Results

### Regularisation on simulated data of Michaelis-Menten kinetics

Michealis-Menten UDE models were trained for different regularisation strengths, and sampling schedules. Figure 2 visualises 25 model fits with the lowest training error for models trained both with ($\lambda = 10^{-5}$ and $\lambda = 1$) and without regularisation for a sampling duration of 40 minutes with samples taken every 5 minutes. The grey area indicates where data has been supplied for parameter estimation. Comparing the model fits to the ground truth model (black, solid and dashed), we see that in the regularised cases (B) and (C) there is good correspondence between the model simulation and the ground truth throughout the 100 minutes, while the unregularised models (A) fail to predict the concentrations correctly beyond the supplied data. Furthermore, the interquartile ranges are narrower for the models trained with regularisation indicating a lower variance in model prediction. Differences between regularisation strengths are also observed. The correspondence between model and simulated true values is higher for $\lambda = 10^{-5}$ than for $\lambda = 1$. We also compared model training without regularisation to regularisation with $\lambda = 10^{-5}$ using a simulated dataset with 10% noise, of which the forecasts can be seen in S1 Fig. For edge cases, either when the amount of available data

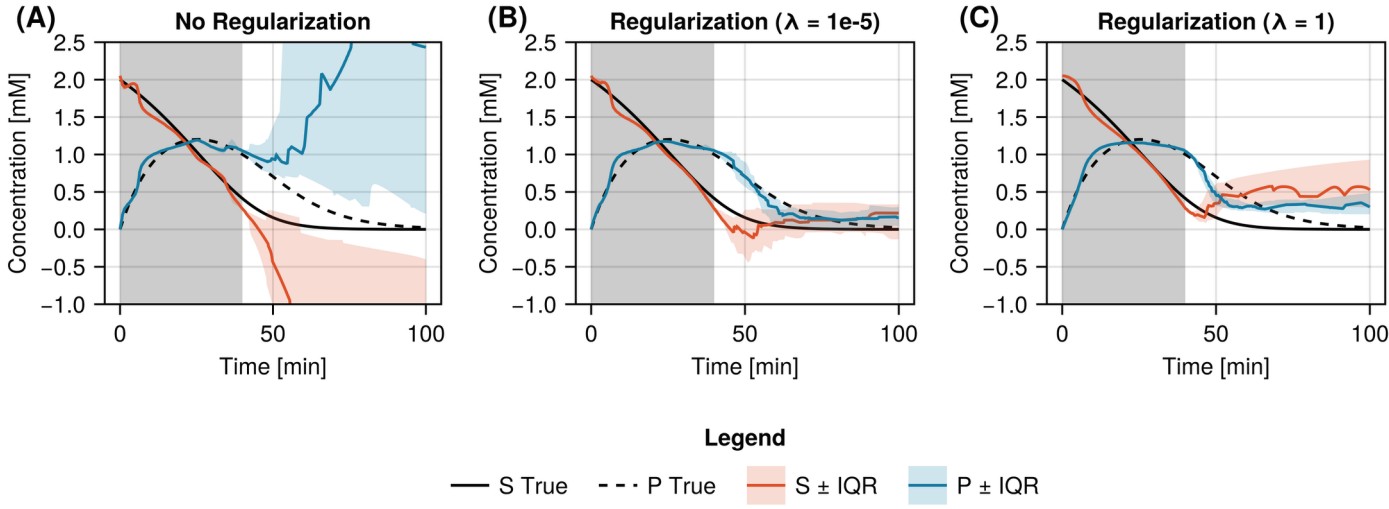

**Fig 2. Visualisation of learned Michaelis-Menten models trained with and without regularisation.** Median predictions for the Michaelis-Menten model for species S (red) and species P (blue) including the first and third quartiles for A: the case without regularisation ($\lambda = 0$), B: mild regularisation ($\lambda = 10^{-5}$), and C: strong regularisation ($\lambda = 1$) for a sampling duration of 40 minutes. The red and blue shaded regions indicated the interquartile range of the top 25 models, selected based on the training error, for species S and P respectively. The sampling duration used for training is marked in the grey shaded region. The ground truth model is also visualised with the black solid (S) and dashed (P) lines to allow comparison of the model fits.

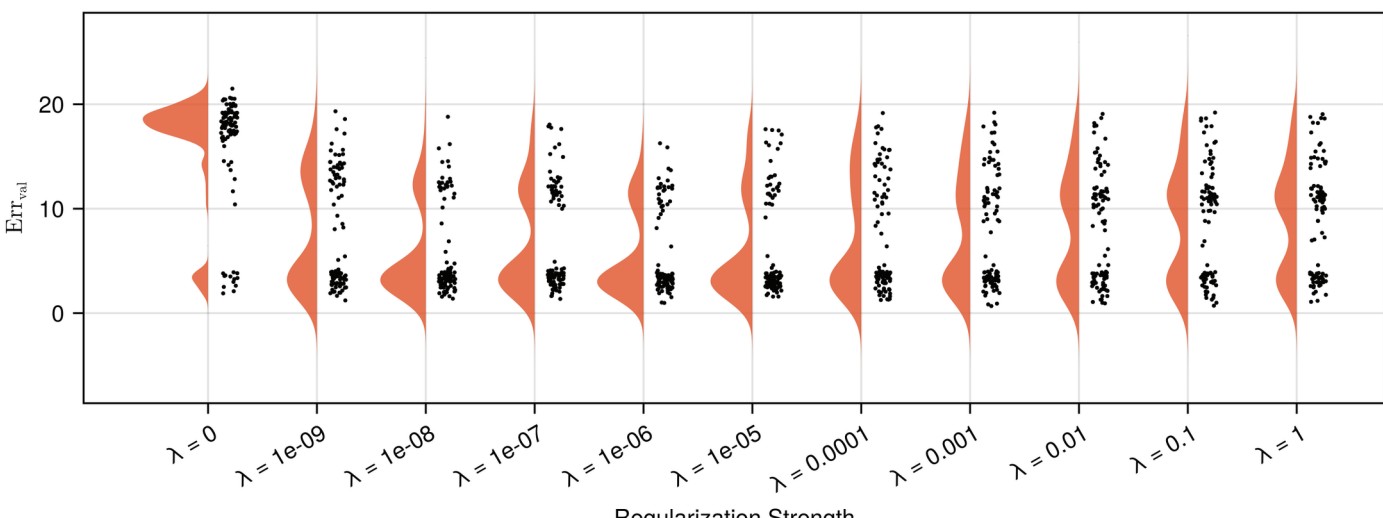

**Fig 3. Distributions of the validation error for Michealis-Menten models trained with different regularisation strengths.** The distributions for of validations errors ($Err_{val}$) computed on each of the 100 models for different regularisation strengths. The models are trained using data over a sampling duration of 40 minutes. $Err_{val}$ are computed on the simulated ground-truth data over a duration of 400 minutes. The validation error is presented on a log10 scale. For models trained without regularisation ($\lambda = 0$) a biphasic distribution in validation errors is found, with the majority of model fits finding local minima with a large $Err_{val}$. The incorporation of regularisation ($\lambda \geq 10^{-9}$) during the training of the Michaelis-Menten UDE improves the long-term dynamics of the learned model terms with the majority of validation errors located in the vicinity of $Err_{val} = 4$. The effect of the regularisation strength on the performance is mainly seen in the size of the peak around $Err_{val} = 4$, which becomes larger for stronger until $\lambda = 10^{-5}$ and then flattens out, as more model fits remain at larger validation error values.

is much smaller (S3 Fig), or when there is sufficient data for the model to learn the behaviour without regularisation (S4 Fig), the benefit of regularisation is limited.

Additionally, we investigated the distributions of the ground-truth evaluation error ($Err_{val}$) for these initialisations, which are shown in Figure 3. In this case, the used training set was sampled every five minutes for 40 units of time. The figure shows a clear bimodal distribution when the models are trained without regularisation, with the majority of model fits finding an erroneous steady state with a large $Err_{val}$, whereas for all regularisation strengths the majority of the model initialisations have a low validation error, as evident with the peak close to an $Err_{val}$ of four. The distributions of all regularised cases are similar in location, but differ slightly in shape. For increasing regularisation strengths, the peak around $Err_{val} = 4$ becomes sharper until $\lambda$ becomes larger than $10^{-5}$, after which the peak flattens out, indicating a reduced performance improvement for larger $\lambda$-values. An additional validation of the benefit of regularisation on a more complicated model is included in S1 Text.

We also evaluated the effect of data sampling duration and sampling sparsity on the quality of the learned model. Figure 4 shows the mean evaluation errors computed for the unregularised case (orange) and two regularised cases ($\lambda = 10^{-5}$ and $\lambda = 1$) (blue and green respectively). The sampling duration varied from 20 to 100 minutes in steps of 10 minutes for samples taken every 5 and 10 minutes. For samples taken every 20 minutes, the sampling duration varied from 40 to 100 minutes in steps of 20 minutes.

The values in Figure 4 show that the average model reconstruction error in the regularised cases (blue and green) do not seem strongly influenced by the sampling sparsity, whereas the increase in sampling duration reduces the validation error. For slight regularisation, we observe the lowest validation error, consistently outperforming the average unregularised

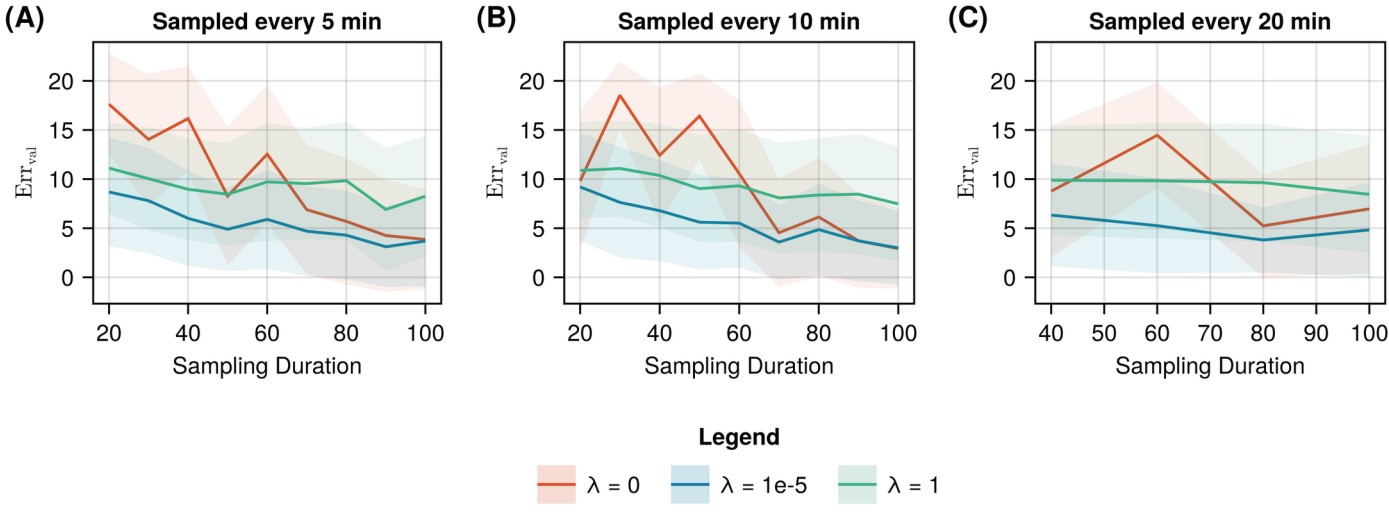

**Fig 4. Mean validation error for varying durations of data collection and sampling frequencies with and without regularisation.** Mean (solid line) and standard deviation (shaded) of the validation error of the optimised models with slight ($\lambda = 10^{-5}$), strong ($\lambda = 1$), and without regularisation, for different sampling durations of the training data sampled every A: 5 minutes, B: 10 minutes, and C: 20 minutes.

models up until a sampling duration of 80 minutes, after which the average performance of both conditions remains similar. For the unregularised case (red) we observe a higher validation error than the regularised case until a sample duration of 60 minutes, after which the unregularised case shows comparable validation errors. For longer sampling durations, the difference between regularised and unregularised models becomes small (S5 and S6 Figs). We additonally evaluated the robustness of the trained UDE to extrapolate to other conditions by simulating the trained model for a different initial condition (S2 Fig).

### Effect of physiology-based regularisation on human data

To evaluate the generalisability of the benefit of physiology-based regularisation on the training of UDE models, we also applied the approach in a second example using human data. In this instance, to enable simulation of the glucose response to a complex meal we aimed to use a universal approximator differential equation to learn an adjusted glucose rate-of-appearance term in the glucose minimal model.

Hyperparameter values for both non-negativity and area-under-curve regularisation were evaluated. The fits of the top 25 models based on training error for models train with ($\lambda_{nonneg} = 100$ and $\lambda_{AUC} = 1$) and without regularisation are shown in Figure 5A and 5B. In both conditions the models can be seen to fit the average measured data. Moreover, following the meal, the models trained with regularisation tend to converge to a steady-state approximately equal to the fasting plasma glucose concentration measured prior to the meal. Whereas the models trained without regularisation either find no steady-state or a steady-state within ± 0.5 mmol/l of the fasting plasma glucose. In Figure 5C and 5D, the rate-of-appearance curves for each of the models are shown. Both curves show a biphasic glucose appearance. At the start of both curves, the regularised curve is consistently positive, while some unregularised curves become negative. Furthermore, the observed difference in steady-state in the plasma glucose curves between the regularised and unregularised cases can also

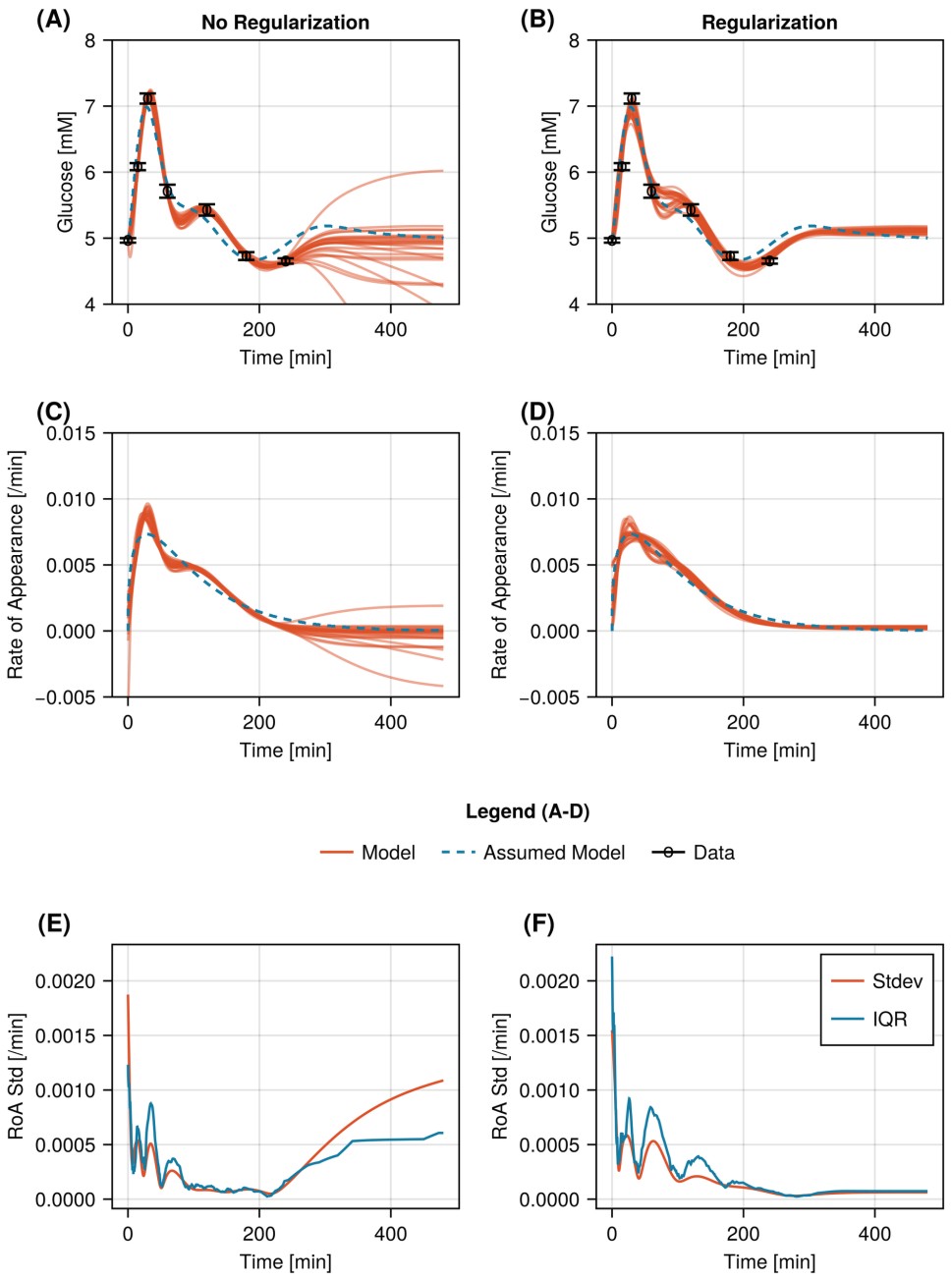

**Fig 5. Results of the 25 glucose-minimal UDEs with the lowest training error values for models trained with and without regularisation.** A: Fits of models trained without regularisation (red) and the original rate-of-appearace (blue dashed) to measured plasma glucose concentrations, where data is shown as mean (black circles) ± standard error. B: Fits of models trained with regularisation (red) and the original rate-of-appearace (blue dashed) to measured plasma glucose concentrations, where data is shown as mean (black circles) ± standard error. C: Learned rate-of-appearance functions for the 25 glucose-minimal UDEs with the lowest training error values trained without regularisation (red) and the original rate-of-appearance function (blue dashed). D: Learned rate-of-appearance functions for the 25 glucose-minimal UDEs with the lowest training error values trained with regularisation (red) and the original rate-of-appearance function (blue dashed). E: Standard deviation (red) and interquartile range (blue) of the rate-of-appearance function over time for the 25 unregularised models with the lowest training error values. F: Standard deviation (red) and interquartile range (blue) of the rate-of-appearance function over time for the 25 regularised models with the lowest training error values.

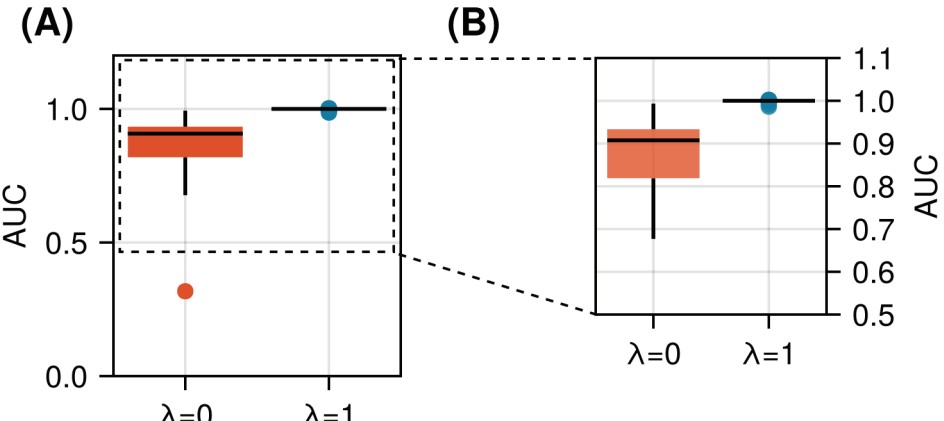

**Fig 6. Area under the rate-of-appearance curve for the regularised and unregularised models.** A: Distribution of the area under the curve of the unregularised (red) and regularised (blue) models. B: Detailed view of the top part of the figure in A.

be seen in the rate-of-appearance curves. In Figure 5E and 5F, the standard deviation and interquartile ranges of the rate-of-appearance curves are visualised over time, demonstrating that physiology-based regularisation improves long term stability of the learned solution as the steady state standard deviation and interquartile range remain low. In the unregularised case, both metrics increase after the final glucose data point. The effects of the individual regularisation terms are shown in S7 Fig.

To verify that regularisation on the area under the curve results in conservation of glucose mass, the AUC of the learned rate-of-appearance curves was computed for models trained both with and without regularisation. Figure 6 shows that the regularisation strongly forces the AUC of the rate of glucose appearance to 1.0, with some models showing an AUC between 0.9 and 1.0, while the spread of the unregularised AUC values is markedly larger.

## Discussion

While methods combining machine learning with knowledge-based models, such as UDEs, have great potential to rapidly advance modelling in systems biology, the increased model complexity coupled with the number of trainable parameters still limits their use to either well-studied systems, or cases where abundant data is readily available. In this study, we demonstrate the benefit of physiology-informed regularisation as a technique to both stabilise training of UDEs for biological systems, as well as increase the biological plausibility of the found solutions.

We have demonstrated the advantage of regularisation techniques in a simulated example, where the ground truth was available. In the simulated case, we show that physiology-informed regularisation supports successful training of a UDE model in low data scenarios. Furthermore, distributions of validation errors from different initialisations indicated that regularisation increases the likelihood of learning more accurate models. Additionally, we have demonstrated that the benefits of physiology-informed regularisation are greatest with shorter sampling durations, reflecting the ability of the regularised model to fit the system with less information. However, we also show the necessity for hyperparameter tuning,

as a too strong regularisation can negatively impact the forecasting and generalisation abilities of the model. We demonstrate that incorporating physiology-informed regularisation shows benefit when applied on real-world data, by applying it to learn a representation of the rate of glucose appearance from a meal in the glucose minimal model using meal responses measured in healthy humans.

In this study, we trained 100 models for each scenario using random parameter initialisation. We used the training error to construct a population of top performing models. When using regularisation during model training, the population's capacity to predict behaviours beyond the training data consistently outperformed the population of models trained without regularisation (Figs 2, 5A and 5B). Particularly, the UDE model without regularisation, seemed to perform better at longer times between samples, especially combined with low sampling durations. However, this result as observed in Figure 4, results from the fact that the model trained on data that was sampled every 10 or 20 minutes learned a steady-state (S3 Fig for an example), while the model trained on data sampled every 5 minutes did not, resulting in equally unwanted behaviour, but differences in model error. Nevertheless, in both simulated and real-life situation regularisation was shown to reduce the variability in found solutions with the lowest training error. In the real-life situation, this effect is smaller than in the simulated case. Furthermore, based on the differences between the standard deviation and the interquartile range in the real-life situation it is likely that the continued increase in standard deviation is driven mostly by outlier model fits, while the eventual stabilisation of the increased interquartile range reflects the greater distribution in learned steady-state values. The effects of the individual regularisation terms can be summarised from S7 Fig. The non-negativity penalty reduces downward trends in model behaviour, after the final timepoint in the data, whereas the area under the curve penalty further stabilises the model simulations. An additional benefit of the combination of regularisation terms is that the nonnegativity penalty prevents the model from compensating for the area under the curve penalty by predicting a negative rate of appearance.

This work extends earlier applications of physiology-informed regularisation [3,15] into the domain of scientific machine learning, and provides a simple alternative to earlier methods to improve UDE stability during training [11,13,14]. As basic physiological or chemical constraints are often available in biological models, the extension of a UDE training process with physiology-informed regularisation is often straightforward. While previous methods of improving UDE training have shown their efficacy in other engineering fields, the limited data availability often encountered in biological applications makes the application of these methods challenging. By accounting for straightforward biological constraints, the physiology-based regularisation approach presented in this work addresses the training stability by providing additional information about the system, as well as guiding the gradient of the cost function towards biologically more plausible solutions.

Despite the higher performance of UDE models when combined with regularisation, it should be noted that a universal differential equation remains data-driven, and its successful application heavily relies on sufficient data availability. It has been shown that this hybrid approach combining ML with mechanistic models can be trained with fewer learning examples than pure ML algorithms [7]. However, with sparse data they are still prone to overfitting. In this work, we show that the incorporation of physiology-informed regularisation can aid in the training of UDEs on sparse and noisy data. In particular, we show that for shorter sample durations or very sparsely sampled data, the incorporation of additional knowledge in the form of physiology-informed regularisation can aid in the learning of biologically plausible models. Furthermore, if the data becomes too sparse, noisy, or if data is missing over highly dynamic regions of the system behavior, the UDEs will be unable to recover

this behaviour, like any machine learning model. Whereas a UDE is computationally more demanding compared to a regular ordinary differential equation model, the size of the neural network that the available data allows for is typically limited. S1 Text additionally shows that the regularisation benefit is not limited to simple kinetic terms, but can also be applied in slightly larger, and more complicated models. However, evaluations on larger systems than the models presented here are still warranted. Therefore, components of validated ODE models should be preferred over neural network terms, which both increases the computational efficiency, but more importantly allow for the data to be used most effectively in parts of the model where biological knowledge is limited.

Another method that bears resemblance to the type of regularisation demonstrated here, is ridge regression, which instead of constraining model behavior, pulls parameters towards predetermined values during model training and is often used in purely data-driven machine learning applications to limit model overfitting on uninformative features by penalising the size of the parameters directly. However, in context of neural networks, the individual parameters have limited meaning by themselves. While a form of ridge regression can still be useful in UDE training to prevent strong overfitting on the data, this does not prevent the model from learning biologically implausible behaviour. Nevertheless, if necessary, both ridge regression and physiology-informed regularisation could also be used in conjunction, noting of course that this would introduce an additional hyperparameter to be tuned.

While the results indicate clear benefits of regularisation for training of UDE models for biological systems, all model fits in the real-life scenario were performed on the average glycaemic response. We have not tested regularisation in a setting where models were fit to individual meal responses, as the stronger contribution of measurement noise to that type of data complicates solid assessment of regularisation techniques. Furthermore, we have fixed the parameters of the meal response model to the values estimated using the hypothetical rate-of-appearance function. As a result, we may bias the learned neural rate-of-appearance towards a specific shape generated by the initial glucose minimal model. However, this limitation is present in both regularised and unregularised models trained here. In the future, this could potentially be investigated in further research in context of the capabilities of neural networks within UDE models to correct for errors in model parameters outside of the neural network. Furthermore, while the inclusion of this neural network into the system provides satisfactory forecasting abilities, the interpretability of the model is reduced. To improve interpretability, tools like symbolic regression [34,35] could be applied after training, to derive a functional representation of the missing model component [7].

Additionally, while we have demonstrated the benefits of physiology-informed regularisation in both a simulated and real data setting, the models evaluated in this work are relatively simple, describing interactions between two state variables. In future work, the efficacy of physiology-informed regularisation should be confirmed in larger model systems.

While in this manuscript we focus on UDEs, where the neural network is directly embedded into a systems of differential equations, Physics-Informed [19] (PINNs), or Systems Biology-Informed [36], neural networks (SBINNs), have emerged as an alternative approach to combining neural networks with systems of differential equations. In PINNs, the neural network is used to directly learn the dynamic behaviour of the system, where it is regularised by a known system of differential equations, such that the found solution adheres to known physical laws. However, previous research on PINNs and SBINNs has shown these to be sensitive to overfitting, particularly in applications with sparse data [37]. The physiology-informed regularisation presented in this study may also be beneficial for the training of PINNs or SBINNs on less frequently sampled data, however this would warrant further investigation.

In summary, we have presented physiology-informed regularisation as a simple, yet powerful and generalisable approach for the improvement of UDE training in biological models. The use of physiology-informed regularisation not only improves long-term predictive stability, reducing model variance, but it can also be seen to reduce non-physiological behavior in the neural network component, facilitating the training of UDEs on sparse data.

## Author contributions

**Conceptualization:** Max de Rooij, Balázs Erdős, Natal A. W. van Riel, Shauna D. O'Donovan.

**Formal analysis:** Max de Rooij.

**Funding acquisition:** Natal A. W. van Riel, Shauna D. O'Donovan.

**Investigation:** Max de Rooij.

**Methodology:** Max de Rooij.

**Software:** Max de Rooij.

**Supervision:** Natal A. W. van Riel, Shauna D. O'Donovan.

**Visualization:** Max de Rooij.

**Writing – original draft:** Max de Rooij.

**Writing – review & editing:** Balázs Erdős, Natal A. W. van Riel, Shauna D. O'Donovan.

## Supporting information

**S1 Fig. Visualisation of learned Michealis-Menten models trained with and without regularisation on data with 10% noise.** Median predictions for the Michealis-Menten model for species S (red) and species P (blue) including the first and third quartiles for A: the case without regularisation ($\lambda = 0$), and B: mild regularisation ($\lambda = 10^{-5}$) for a sampling duration of 40 minutes. The red and blue shaded regions indicate the interquartile range of the top 25 models, selected based on the training error, for species S and P respectively. The sampling duration used for training is marked in the grey shaded region. The ground truth model is also visualised with the black solid (S) and dashed (P) lines to allow comparison of the model fits. (TIF)

**S2 Fig. Visualisation of learned Michealis-Menten models trained with and without regularisation on a modified initial condition** Median predictions for the Michealis-Menten model for species S (red) and species P (blue) including the first and third quartiles for A: the case without regularisation ($\lambda = 0$), B: mild regularisation ($\lambda = 10^{-5}$), and C: strong regularisation ($\lambda = 1$) for a sampling duration of 40 minutes. The red and blue shaded regions indicate the interquartile range of the top 25 models, selected based on the training error, for species S and P respectively. The sampling duration used for training is marked in the grey shaded region. The ground truth model is also visualised with the black solid (S) and dashed (P) lines to allow comparison of the model fits. (TIF)

**S3 Fig. Visualisation of learned Michealis-Menten models trained with and without regularisation, on a sampling duration of 20 minutes, sampled every 10 minutes.** Median predictions for the Michealis-Menten model for species S (red) and species P (blue) including the first and third quartiles for A: the case without regularisation ($\lambda = 0$), B: mild regularisation ($\lambda = 10^{-5}$), and C: strong regularisation ($\lambda = 1$) for a sampling duration of 20 minutes. The red

and blue shaded regions indicate the interquartile range of the top 25 models, selected based on the training error, for species S and P respectively. The sampling duration used for training is marked in the grey shaded region. The ground truth model is also visualised with the black solid (S) and dashed (P) lines to allow comparison of the model fits.
(TIF)

**S4 Fig. Visualisation of learned Michealis-Menten models trained with and without regularisation, on a sampling duration of 80 minutes, sampled every 5 minutes.** Median predictions for the Michealis-Menten model for species S (red) and species P (blue) including the first and third quartiles for A: the case without regularisation ($\lambda = 0$), B: mild regularisation ($\lambda = 10^{-5}$), and C: strong regularisation ($\lambda = 1$) for a sampling duration of 80 minutes. The red and blue shaded regions indicate the interquartile range of the top 25 models, selected based on the training error, for species S and P respectively. The sampling duration used for training is marked in the grey shaded region. The ground truth model is also visualised with the black solid (S) and dashed (P) lines to allow comparison of the model fits.
(TIF)

**S5 Fig. Visualisation of learned Michealis-Menten models trained with and without regularisation, on a sampling duration of 200 minutes, sampled every 5 minutes.** Median predictions for the Michealis-Menten model for species S (red) and species P (blue) including the first and third quartiles for A: the case without regularisation ($\lambda = 0$), B: mild regularisation ($\lambda = 10^{-5}$), and C: strong regularisation ($\lambda = 1$) for a sampling duration of 200 minutes. The red and blue shaded regions indicate the interquartile range of the top 25 models, selected based on the training error, for species S and P respectively. The sampling duration used for training is marked in the grey shaded region. The ground truth model is also visualised with the black solid (S) and dashed (P) lines to allow comparison of the model fits.
(TIF)

**S6 Fig. Visualisation of learned Michealis-Menten models trained with and without regularisation, on a sampling duration of 400 minutes, sampled every 5 minutes.** Median predictions for the Michealis-Menten model for species S (red) and species P (blue) including the first and third quartiles for A: the case without regularisation ($\lambda = 0$), B: mild regularisation ($\lambda = 10^{-5}$), and C: strong regularisation ($\lambda = 1$) for a sampling duration of 400 minutes. The red and blue shaded regions indicate the interquartile range of the top 25 models, selected based on the training error, for species S and P respectively. The sampling duration used for training is marked in the grey shaded region. The ground truth model is also visualised with the black solid (S) and dashed (P) lines to allow comparison of the model fits.
(TIF)

**S7 Fig. Model fits of the glucose minimal model including the neural network rate-of-appearance term for varying regularisation strengths.** Each solid line represents a model with a unique initial parameter set. The data is shown in circles. Each column represents the non-negativity regularisation strength, while each row represents the area-under-curve regularisation strength.
(TIF)

**S1 Text. Regularisation UDE in an extended Michaelis-Menten model with competitive inhibition.** As an additional validation of the regularisation performance, the Michaelis-Menten model was extended by including an additional inhibitor molecule *I*. Using the more complicated model, the benefit of physiology-informed regularisation was evaluated.
(PDF)

**S2 Text. Mathematical description of an assumed rate-of-appearance function used to estimate the model parameters of the glucose minimal model.** As a rate-of-appearance function, a delayed pulse input was modelled, described by Korsbo et al. [33]. (PDF)

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
