## [Decision Letter · Decision Letter 0]

4 Sep 2024

Dear mr de Rooij,

Thank you very much for submitting your manuscript "Physiology-informed regularization enables training of universal differential equation systems for biological applications" for consideration at PLOS Computational Biology.

As with all papers reviewed by the journal, your manuscript was reviewed by members of the editorial board and by several independent reviewers. In light of the reviews (below this email), we would like to invite the resubmission of a significantly-revised version that takes into account the reviewers' comments.

We cannot make any decision about publication until we have seen the revised manuscript and your response to the reviewers' comments. Your revised manuscript is also likely to be sent to reviewers for further evaluation.

Sincerely,

Varun Dutt, Ph.D

Academic Editor

PLOS Computational Biology

Christoph Kaleta

Section Editor

PLOS Computational Biology

Reviewer's Responses to Questions

**Comments to the Authors:**

Reviewer #1: The manuscript presents and evaluates a method to regularize universal differential equations (UDE) to avoid or at least limit unrealistic solutions. The approach is tested using two well-known examples in Systems Biology: the case of an enzyme catalysed reaction with Michaelis-Menten kinetics and blood glucose levels following ingestion of a meal.

Appearance of negative concentrations is a well known issue of ODE's when used in Systems Biology and the proposed approach appears to limit those non-biological/non-realistic cases very efficiently when the approach is extended to UDEs. In the case of blood glucose levels the regularization approach also seems to accomplish the fact that all ingested glucose eventually appears in blood (and does not disappear in the gastrointestinal tract).

Universal differential equations are a promising approach to combine the advantages of knowledge based models built using differential equations and the potential of machine learning (neural networks). This method tests, illustrates and expands this work and it is a useful addition to the field.

A number of issues should be addressed

Code and code availability and accessibility: the code has been made available in git. The code is reasonably well structured and it is possible to run the examples. Dependencies are not clearly stated in the Readme file. Please list the required dependencies for each example so that the code can be tested.

The code is not commented, please add comments so that the code can be easily read, understood and adapted. Add descriptive comments to the functions to clarify what the inputs and outpus are. Also add inline comments to describe what the code is intended to produce and what the variables denote. For example, reading the function michaelismenten_ude it is not directly evident to know which A or B is the product or the substrate.

Materials and Methods:

Notation and explanations of the equations can largely be improved by using more accurate and concise mathematical notation. The vector variables are usually denoted with bold font (or and “arrow “ on top of the variable). This is done here inconsistently. For example in equation 1 it is clear that u denotes a vector, but the p and \theta (sets of parameters) are not considered vectors but scalars. Also equations 2 and 3 use bold font to denote S and P (which are scalars…). Similarly G and X in equations 8 and 9 should not be in bold font. Please check everywhere and ensure consistent use of bold to denote vectors.

Also from equations 2 and 3 it appears that a vector u is defined that is used in later on (equations 5 to 7).. however this vector is never explicitly defined. Finally the use of A and B as state variables line 149 appears to be a mistake that should be corrected (even though it does link with the code files).

The lambda values provided in line 166 appear incomplete, as in the text a lambda value of 1 is often cited, (that should match to x=0, not included in the range).

Please include in the text the regularization penalty that is included to ensure the glucose area under the curve is one. This is an important function that should be explicitly included.

Please double check the precision and accuracy of the mathematical descriptions in the materials and methods section.

Results

The authors compared the performance of the UDE system with and without the regularization function. However they do not compare to an ODE only model in both cases. The authors should briefly comment in which cases using UDE outperforms the use of ODE models for the applications here presented.

The models are set to be simulated in real life settings with 5% noise. To what extent is this 5% a reasonable representation of the noise in these systems? How would the approach work with higher ( 10% , 15%) noise levels.

Also a typical characteristic of the considered problems is the appearance of missing data points. How would the method perform if data points are missing from the time series?

Figures 2 and 3 provide results corresponding to a sampling duration of 40 min every 5min. To what extent are the results relevant for other sampling scenarios? In addition to the data in Figure 4 the authors should indicate whether the observed improvements also hold for more extreme sampling cases.

Figure 3 legend: why is it stated (\lambda > 0.1 as a criterion for there being a regularization? Is this a typo?

Figure 4. When comparing the effect of different sampling frequencies it appears that sampling every 20 minutes leads to lower error than sampling every 5 or 10 minutes (at least for lambda=0). Can the authors comment on this? Also what happens if very long sampling durations are considered (up to 400 minutes? )

Figure 5 shows a clear difference between the regularized and the non regularized models. From the two regularization terms introduced (negative concentrations, area under the curve) what is the impact of each one? Is this effect due to the combination of both terms or can it be attributed to only one of them? From 5a and 5c it seems that the appearance of negative G/X is not happening often, so that regularization should not have a major effect. Can the authors evaluate the relative impact of each contribution?

Typos and Mistakes: please check the spelling of Michaelis-Menten (there are multiple cases of Michealis-Menten) that should always be written with upper case letters. (Same for Gaussian )

Reviewer #2: Regularization methods for generalizing Neural Networks for learning UDE parameters with real and simulated data are presented by the authors. Although, the approach and results presented are limited to a simple two state model, details of the approach and limitations are well documented

A minor change:

Line 374: In summary, we have presented physiology-informed regularization as a simple, yet powerful and generalizable approach for the improvement of UDE training in biological models. The used of physiology-informed regularization not only improves long-term predictive stability, reducing model variance, but it can also be seen to reduce non-physiological behavior in the neural network component.

Reviewer #3: Comments are attached.

**Have the authors made all data and (if applicable) computational code underlying the findings in their manuscript fully available?**

Reviewer #1: Yes

Reviewer #2: None

Reviewer #3: Yes

PLOS authors have the option to publish the peer review history of their article (what does this mean?). If published, this will include your full peer review and any attached files.

Reviewer #1: **Yes: **Maria Suarez Diez

Reviewer #2: No

Reviewer #3: No
---

## [Decision Letter · Decision Letter 1]

6 Jan 2025

Dear mr de Rooij,

We are pleased to inform you that your manuscript 'Physiology-informed regularization enables training of universal differential equation systems for biological applications' has been provisionally accepted for publication in PLOS Computational Biology.

Best regards,

Varun Dutt, Ph.D

Academic Editor

PLOS Computational Biology

Christoph Kaleta

Section Editor

PLOS Computational Biology

Reviewer's Responses to Questions

**Comments to the Authors:**

Reviewer #1: My comments have been addressed

**Have the authors made all data and (if applicable) computational code underlying the findings in their manuscript fully available?**

Reviewer #1: Yes

PLOS authors have the option to publish the peer review history of their article (what does this mean?). If published, this will include your full peer review and any attached files.

Reviewer #1: **Yes: **Maria Suarez-Diez

---

## [Editor Report · Acceptance letter]

PCOMPBIOL-D-24-00878R1

Physiology-informed regularization enables training of universal differential equation systems for biological applications

Dear Dr de Rooij,

I am pleased to inform you that your manuscript has been formally accepted for publication in PLOS Computational Biology. Your manuscript is now with our production department and you will be notified of the publication date in due course.

With kind regards,

Zsofia Freund
